# Identification and Migration Studies of Photolytic Decomposition Products of UV-Photoinitiators in Food Packaging

**DOI:** 10.3390/molecules24193592

**Published:** 2019-10-06

**Authors:** Joseph B. Scarsella, Nan Zhang, Thomas G. Hartman

**Affiliations:** Department of Food Science, Rutgers, The State University of New Jersey, 65 Dudley Road, New Brunswick, NJ 08901, USA; nz81@scarltemail.rutgers.edu

**Keywords:** photoinitiator, energy-curable, food packaging migration, photolytic decomposition product, NIAS, gas chromatography-mass spectrometry, electrospray-mass spectrometry

## Abstract

UV-curable inks, coatings, and adhesives are being increasingly used in food packaging systems. When exposed to UV energy, UV-photoinitiators (PI’s) present in the formulations produce free radicals which catalyze polymerization of monomers and pre-polymers into resins. In addition to photopolymerization, other free radical reactions occur in these systems resulting in the formation of chemically varied photolytic decomposition products, many of which are low molecular weight chemical species with high migration potential. This research conducted model experiments in which 24 commonly used PI’s were exposed to UV-energy at the typical upper limit of commercial UV-printing press conditions. UV-irradiated PI’s were analyzed by gas chromatography-mass spectrometry (GC-MS) and electrospray-mass spectrometry (ESI-MS) in order to identify photolytic decomposition products. Subsequently, migration studies of 258 UV-cure food packaging samples were conducted using GC-MS; PI’s and photolytic decomposition products were found in nearly all samples analyzed. One hundred-thirteen photolytic decomposition products were identified. Eighteen intact PI’s and 21 photolytic decomposition products were observed as migrants from the 258 samples analyzed, and these were evaluated for frequency of occurrence and migratory concentration range. The most commonly observed PI’s were 2-hydroxy-2-methylpropiophenone and benzophenone. The most commonly observed photolytic decomposition products were 2,4,6-trimethylbenzaldehyde and 1-phenyl-2-butanone. This compilation of PI photolytic decomposition data and associated migration data will aid industry in identifying and tracing non-intentionally added substances (NIAS) in food packaging materials.

## 1. Introduction

Food packaging is ubiquitous in modern society. Nearly all food products sold in stores come pre-packaged in multi-functional packages designed to protect food from damage and to elongate food shelf-life. Food packages usually are printed with inks and coatings to attract attention from potential customers and convey information about the product enclosed therein. While inks are essential in printing colorful graphics onto food packaging, coatings are used to add additional gloss or matte finish, increase ink abrasion resistance and even improve a package’s resistance to moisture or gas permeation [1]. Food packages are made from a wide variety of materials including metals, glass, plastics, and paper in order to achieve better functional properties for specific types of food products. Adhesives are commonly used to combine these different materials together into laminate structures, seal food packages, and affix packaging labels. Many food packaging inks, coatings and adhesives employ UV-cure technology which is more environmentally friendly and provides faster line speeds than conventional aqueous and solvent based ink, coating, and adhesive systems. The UV-curable inks, coatings, or adhesives can be cured onto substrates through a free-radical polymerization reaction induced by light sensitive chemicals called UV-photoinitiators (PI’s) which absorb energy under ultraviolet (UV) light and generate free radicals either on their own or through interaction with a co-initiator [2]. Even though UV-curable systems are widely used and promoted in the food packaging industry, there are concerns about unreacted residual components of UV materials such as PI’s and other monomers [3]. These components generally have low molecular weight and can migrate into food systems causing safety risks [4]. Typical UV inks, coatings, and adhesives employ PI mixtures which can generate large excess of free radicals to improve cure speed and counter oxygen inhibition. Excess free radicals generated from UV cure process will be involved in further photolytic decomposition, rearrangement and recombination reactions which can generate new chemical species [5]. These photolytic decomposition products are also highly migratory because of their low molecular weight, and this brings forth additional food safety concerns. Migration of volatile and odorous compounds such as photolytic decomposition products of PI’s can negatively influence food product safety as well as sensory attributes and consumer acceptance of food products [6].

Between 2000 and 2011, 143 notifications were issued by Rapid Alert System for Food and Feed (RASFF) in the European Union regarding the migration of PI’s from food packaging into foodstuffs. In September 2005, an Italian RASFF notification reported observed migration of the PI 2-isopropylthioxanthone (ITX) at 250 µg/L into packaged infant formula, leading to the recall of over 30 million liters of infant formula throughout Europe. In 2009, ten RASFF notifications were issued regarding the migration of the PI’s benzophenone and 4-methylbenzophenone. These notifications led to the recall of several batches of breakfast cereals and nearly 7 tons of milk across Europe [7]. Stimulated by the increasing instances of noted food contamination from PI’s, many countries began establishing regulations regarding allowable migration levels of PI’s in food packaging systems; international organizations and corporations like European Printing Ink Association and Nestle also start to set related standards and limitations [8,9]. In March 2008, the Swiss Federal Department of Home Affairs (FDHA) adopted an amendment to the Ordinance of Foodstuffs and Utility Articles of 2005 which detailed regulations for ink components in food packaging systems. The ordinance has listed specific migration limits (SML) for 24 evaluated PI’s [10]. European Commission and German Institute of Risk Assessment (BfR) were also working on similar legislations on food packaging ink components. In United States, state laws like California Proposition 65 especially listed benzophenone, a very commonly used PI, as a carcinogen [11]. FDA also removed benzophenone (listed as synthetic flavoring substance and plastic additive) from approved food additives lists in 2018 [12]. Despite the regulations on PI’s used in food packaging, few regulations cover photolytic decomposition products of PI’s migrating into food products from packaging materials. The photolytic decomposition products fall into the category of non-intentionally added substances (NIAS) given in EU Commission Regulation No. 10/2011 [13]. NIAS are unintended food packaging substances which mainly come from impurities, decomposition products, or by-products of the food contact materials [14]. Due to their numerous sources and broad chemical makeup, they are very challenging for regulatory authorities and industrial manufacturers. Two aspects make the identification and quantification of photolytic decomposition products of PI’s particularly challenging. First, there is a scarcity in published literature discussing identification of photolytic decomposition products from PI’s and their migration levels in food packaging materials. This makes it difficult for manufacturers to find the source of photolytic decomposition products and control their formation during food packaging production. Secondly, some photolytic decomposition products have novel structures, and chemical standards often are not available for related quantification and toxicity studies while regulatory authorities require such migration level quantification and toxicity data for evaluation. As a result, toxicological data regarding the safety of many photolytic decomposition products does not presently exist.

In W. Arthur Green’s review of PI chemistry and technology, data were compiled on photolytic decomposition products formed from various PI’s including 2,2-dimethyl-2-hydroxyacetophenone, benzyl dimethyl ketal, 1-hydroxycyclohexylphenylketone, 2-methyl-4′-(methylthio)-2-morpholinopropiophenone, trimethylbenzoyldiphenylphosphine oxide, 2-benzyl-2-(dimethylamino)-4′-morpholinobutyrophenone, ethyl (2,4,6-trimethylbenzoyl) phenylphosphinate, benzophenone and ITX [15]. However, many PI’s commonly used in food packaging materials were not discussed and experimental conditions were not included in this review. Kirschmayr et al. (1982) investigated the generation of yellow products formed during UV-curing of various PI’s. GC-MS analysis of a UV-exposed solution of 1-hydroxycyclohexylphenylketone in methanol identified cyclohexanol, cyclohexanone, benzaldehyde, 2-hydroxycyclohexanone, 1,1-dimethoxycyclohexane, methyl benzoate, 2-hydroxy-1-phenylethanone, 2,2-diphenylethanone, benzoic acid, benzil, and *O*-dibenzoylbenzene as photolytic decomposition products [5]. Nevertheless, only one PI was involved in this research. There are many studies focused on PI migration from food packaging materials, but few studies discuss migration of photolytic decomposition products of PI’s and the frequencies at which they occur as migrants. Lago and Ackerman identified several photolytic decomposition products of PI’s from food packaging using non-targeted GC-MS, UHPLC-MS, and DART (direct analysis in real time)-MS. However, the specific PI’s from which each of the decomposition products were formed were not clearly stated, and migration testing extraction used dichloromethane as a solvent instead of food stimulants which more accurately model migration of real food systems [16].

The current study aims to isolate and identify the photolytic decomposition products formed from PI’s under typically used industrial conditions. This present work will serve as a database of the photolytic decomposition products formed from many PI’s that are commonly used in food packaging materials. Additionally, this research aims to measure the frequencies of occurrence of PI’s and their photolytic decomposition products in food packaging material migration data. Previously published experiments which sought to measure PI migration used several different non-standardized migration test protocols, producing highly variable data. The present research will use standardized migration test protocols aligned with protocols issued by FDA to measure the frequencies of occurrence and the concentration range of both PI’s and their decomposition products.

## 2. Results

### 2.1. Evaluation of Photolytic Decomposition of PI’s

Non-irradiated PI standards were first analyzed by GC-MS to identify impurities, synthesis byproducts, and minor breakdown products present before UV-irradiation resulting from exposure to ambient light. Compounds identified were excluded from results found in UV-irradiated samples. The 24 PI compounds evaluated are listed with structures and CAS Registry numbers in Table 1. Compounds are numbered 1 through 24 and will be referred to by their numbers in the following sections for the sake of conciseness.

A total of 113 photolytic decomposition products were identified, not including tentatively identified oligomers of PI’s. The photolytic decomposition products formed from each PI are listed in Table 2. CAS Registry Number, molecular weight, parent PI’s, peak area percent, and calculated Kovats retention index are included for each compound, where applicable. Products are listed in order of increasing molecular weight. Area percent calculations are based on the total integrated peak area of the parent PI and its photolytic decomposition products.

Table 2 also shows oligomers of PI’s identified by GC-MS. Ions identified from the positive ion mode are protonated molecular ions of dimers and higher oligomers ([2M + H]^+^, [3M + H]^+^, etc.) or sodiated adducts of the molecular ion ([2M + Na]^+^, [3M + Na]^+^, etc.). No ions indicating oligomer formation were detected in negative ion mode experiments.

An exemplar GC-MS total ion current chromatogram of UV-irradiated PI 10 is presented in Figure 1. Numbered peaks correspond to photolytic decomposition products. Figure 2 is an exemplar ESI-MS spectrum obtained from UV-irradiated PI 10 in positive ion scanning mode.

### 2.2. Evaluation of the Occurrence and Migration Level of PI’s and Photolytic Decomposition Products

Data collected from food packaging migration test is presented in Table 3 which shows the calculated frequency of occurrence out of 258 samples for each of 18 PIs and 21 photolytic decomposition products, as well as concentration ranges for each compound separated by both food simulant and FDA condition of use (time and temperature at which testing was conducted). The most commonly observed photolytic decomposition products observed in the migration tests were 2,4,6-trimethylbenzaldehyde, occurring in 130 of 258 samples, 1-phenyl-2-butanone, occurring in 83 of 258 samples, 4-methylthiobenzaldehyde, occurring in 81 of 258 samples, and benzaldehyde, occurring in 78 of 258 samples.

2,4,6-trimethylbenzaldehyde is a decomposition product of PI’s 11, 12, and 13. Experimental data shows that 2,4,6-trimethylbenzaldehyde is formed as a photolytic decomposition product in these PI’s in relatively high concentrations. 2,4,6-trimethylbenzaldehyde comprised 0.996% of the total peak area in GC-MS analysis of UV-exposed PI 13. Migration data shows that this compound is highly migratory at a range of concentration of up to 1938 ng/cm^2^.

1-phenyl-2-butanone is a photolytic decomposition product of PI 8. It is formed in high concentration during UV-exposure; experimental data shows that 1-phenyl-2-butanone comprised 1.05% of the total peak area in GC-MS analysis of UV-exposed PI 8. Migration data shows that 1-phenyl-2-butanone is highly migratory at concentrations up 440 ng/cm^2^.

4-methylthiobenzaldehyde is a photolytic decomposition product of PI 7. The peak for 4-methylthiobenzaldehyde comprised 1.44% of the total peak area in the UV-exposed PI 7 sample. Migration data shows that 4-methylthiobenzaldehyde is a highly migratory compound at concentrations up to 267 ng/cm^2^.

## 3. Discussion

In almost all of the PI’s examined, the intact PI made up the largest peak in the GC-MS chromatogram. Most decomposition products were detected in relatively low concentrations compared to the unreacted PI. In a typical UV-curable ink or coating formulation, free radicals readily and preferentially react with acrylate monomers to build a polymeric chain, and the radical is only quenched by radical recombination when there is no reactive monomer available. In the present experiments, however, no reactant monomer was included in order to demonstrate a worst-case scenario which maximized the formation of photolytic decomposition products. In commercial UV-curable ink, coating, and adhesive formulations, the abundance of acrylate monomer molecules would cause much of the energy from UV-irradiation to be absorbed or to dissipate, causing less energy to reach the PI molecules. Lower rates of radical formation and subsequent photopolymerization and photolytic decomposition reactions would result. Hydrogen donating synergists are often included in UV-curable formulations to counteract this by facilitating the formation of free radicals and the propagation of photopolymerization. Under the present experimental conditions, free radical attack and the formation of photolytic decomposition products occurred as minor side reactions, while the bulk of the parent PI molecules either release energy and return to the ground state from their excited triplet state or radically recombine into the original PI, resulting in a large percentage of the parent compound being maintained [17]. In UV-curable formulations, PI molecules are not consumed by the photopolymerization reaction; PI’s are regenerated once the monomer is depleted, and the low molecular weight molecules remain on the surface and possess high migration potential.

The majority of the photolytic decomposition products identified share general structural features and characteristics to the parent PI. For example, methyldiphenylphosphine oxide, formed upon the decomposition of PI 11, retains the phosphorous-oxygen double bond found in the parent PI. Products such as this were likely formed by UV-induced radical cleavage of various chemical bonds in the PI molecules and recombination of radical fragments in different arrangements. Other products bear little structural resemblance to the parent molecules; these products were likely formed through complex radical formation and recombination reactions in situ. For example, triethylphosphate, identified as a decomposition product of PI 12, bears no similarities with the parent PI other than that it is phosphorous-containing. Formation mechanisms of several selected decomposition products are discussed below, but discussion of individual reaction mechanisms for each product identified is beyond the scope of the present research.

Patterns of photolytic decomposition were quite varied. Some PI’s, such as PI 10, formed a wide range of photolytic decomposition products because the radicals produced upon UV-exposure are extremely reactive, and they are capable of cleaving relatively weak chemical bonds within the PI molecule. Other PI’s, such as PI 14, produced few photolytic decomposition products. Compounds such as this are characteristically highly aromatic and contain bonds weak enough to be cleaved by UV energy or free radicals. Aromatic compounds consist of multiple resonance-stabilized structures, providing enhanced stability and resistance to free radical attacks. The energy from a free radical attack is dispersed among multiple bonds, and the likelihood of bond cleavage occurring is reduced, resulting in the formation of fewer decomposition products from these compounds. When exposed to UV-radiation, PI’s most readily cleave at the bond alpha to the carbonyl group, forming a pair of free radicals. In a usual UV-curable ink, coating, or adhesive formulation, there is sufficient monomer present such that the majority of radicals formed are consumed by photopolymerization of the monomer. However, PI’s are often used in excess in formulations in order to provide rapid cure and to lessen the effects of oxygen inhibition [6]. When excess PI is used, the likelihood of side-reactions including radical recombinations, rearrangements, disproportionation, and reactions with oxygen dramatically increases, all of which result in the quenching of radicals and the formation of new chemical species [3].

Benzaldehyde was a very frequently identified compound, formed as a photolytic decomposition product of PI’s1, 2, 4, 8, and 10. It is usually produced in low concentrations as a result of UV-exposure of PI’s due to the formation of a benzoyl radical. In almost all PI’s, the primary bond cleavage and radical formation occur at the carbon alpha to the carbonyl carbon of the chromophore group, forming a benzoyl radical. Benzaldehyde is formed through the quenching of the benzoyl radical with a free proton. Benzaldehyde is a nearly ubiquitous compound formed as a byproduct of innumerous chemical reactions and is frequently identified in food packaging migration tests. The present data demonstrates one of several potential mechanisms of benzaldehyde formation in food packaging materials. Benzaldehyde could also undergo oxidation by reactive oxygen species such as ozone generated by UV-curing. Short wavelength UV-energy converts diatomic oxygen into ozone, a powerful oxidant [18]. Benzoic acid was identified as a decomposition product of PI’s 1, 2, and 10. Similar aldehydes and their oxidation products were identified as decomposition products of PI’s 7, 11, 12, and 13. Numerous other decomposition products form through recombination of the benzoyl radical or derivatives with other radicals formed from UV-exposure. Acetophenone, for example, identified as a decomposition product of PI’s 1 and 10, was formed though recombination of the benzoyl radical with a methyl radical. Diphenylethanedione, identified as a decomposition product of PI’s 1, 2, and 10, was formed by the recombination of two benzoyl radicals. Similar to benzaldehyde, several other decomposition products were formed as a result of the quenching of a radical species by a proton. These products appear to be simple fragments of the parent PI molecules. For example, *N*,*N*-diethylaniline was formed as a decomposition product of PI 22 in this manner.

Photolytic decomposition products formed through reactions with hydrogen donating synergists, as discussed above, were not investigated in the current research. Hydrogen-donating molecules such as tertiary amines are often added to formulations containing type II PI’s such as benzophenone (PI 14) and methyl 2-benzoylbenzoate (PI 19) to facilitate the formation of free radicals because the energy required to cleave the bond alpha to the chromophore carbonyl group in type II PI’s is greater than the energy provided by UV-light. The synergists form highly reactive radicals upon donating a proton to the PI molecule. An example of a byproduct formed from synergized formulations is the formation of hydroperoxides from benzophenone used in conjunction with methyldiethanolamine as a synergist [19]. Even though it is common practice to use hydrogen donors in UV-curable ink or coating formulations containing type II PI’s, synergists were not used in this study in order to isolate the primary decomposition products of the PI’s without interference. It was expected that the intensity of UV-energy used to prepare the samples would be sufficient to form free radicals from the type PI’s without the need for a synergist. PI’s 22, 23, and 24 contain tertiary amine groups which act as self-synergists, facilitating the formation of free radicals through the same mechanism as discussed above without amines being added to the ink, coating, or adhesive formulation. More numerous decomposition products were observed from PI’s 22 and 23 than most other type II PI’s studied.

ESI-MS was used to supplement GC-MS in the identification of PI oligomers formed as a result of exposure to UV-energy. Although GC-MS detection of oligomers is limited due to volatility, several PI oligomers were identified by GC-MS and are listed in Table 1. Oligomers of PI’s 5, 6, 9, 12, 15, 17, 18, 19, 21, 22, and 23 were tentatively identified by ESI-MS alone. This data shows evidence of radical oligomerization of PI molecules during UV-exposure. However, spontaneous oligomerization of compounds from high energy collisions within an electrospray ion source has been observed in the past. Various molecular and system properties can affect the formation of adducts such as dimers including pK_a_ and surface activity of the analytes and pH, concentration, and the type of solvent system used [20]. The exact structures of the oligomers were not determined.

Although not reported in the present study, ongoing research by our group has shown that benzene is formed at trace levels as a photolytic decomposition product of most UV-cured inks, varnishes, or adhesives (presumably from aromatic ring precursors in the PI’s or other ink, varnish, adhesive, or substrate components). Concentration of benzene production was found to be directly proportional to the amount of UV energy to which the material was exposed during the UV-cure process. Benzene, a highly volatile compound, was not detected in the present study due to evaporative loss during the concentration step of sample preparation. Purge and Trap-Thermal Desorption GC-MS has been used to identify and quantify benzene, using benzene-d_6_ as an internal standard, from various UV-printed food packaging materials containing a broad range of PI’s. Additional research needs to be conducted to thoroughly identify and quantify extremely volatile photolytic decomposition products formed that may be lost using the current analytical methods.

The frequency and rate of migration though a food-contact material of compounds such as PI’s and their decomposition products is dependent on several factors including ink or coating formulation, coating weight, substrate material, molecular size and weight of the compound, and efficiency of the UV-cure process [14]. Physical factors that may influence migration include material storage conditions, lamination and printing conditions, and substrate porosity.

The range of concentrations of a compound extracted into a food simulant under varying conditions can be related to the chemical nature of the compound and its level of use or formation in the sample material. The appropriate food simulant is chosen for a given test in order to most closely mimic the solubility properties of the real food product the packaging is intended to contain. Partition coefficients between substrate and food simulant can vary depending on the solubility of the migrant in the food simulant solvent [21]. Ten percent ethanol food simulant is used to conduct migration tests on packaging intended for aqueous and acidic food products because it will preferentially extract more hydrophilic compounds. Ninety-five percent ethanol food simulant is used to conduct migration tests on packaging intended for fatty food products because it will preferentially extract more lipophilic and hydrophobic compounds. Higher temperatures and longer extraction times tend to increase the concentrations of migrants in the food simulants because molecular diffusion occurs more rapidly at high temperature. It would be expected that a more extreme extraction condition such as FDA Condition of Use “A” (121 °C for 2 h + 40 °C for 10 days) would show higher migrant concentration than a less severe condition such as FDA Condition of Use “C” (66 °C for 2 h + 40 °C for 10 days) [22]. However, this is not the case for many PI’s and decomposition products examined. Other factors such as the frequency of use of certain PI’s, differences in substrate cure efficiency, and migratory tendencies of specific compounds must be considered when examining migration data.

Many food packaging samples tested showed migration of more than one PI and decomposition products of multiple PI’s, including some from PI’s not observed as migrants themselves. It is a common practice for ink and coating manufacturers to employ PI blends in formulations in order to take advantage of desirable properties of multiple PI’s. These properties may include reduced yellowing, low odor, suitability for heavily pigmented materials, increased cure speed, and preferential curing of different pigment colors due to absorption wavelength [23].

It has been observed in many cases that larger, higher molecular weight compounds demonstrate lower migratory tendencies than smaller, lower molecular weight compounds with similar structural features [21]. For this reason, industry has made an effort to shift toward higher molecular weight, more nonvolatile PI’s in applications where migration and odor are primary concerns, such as food packaging [24]. High molecular weight monomeric PI’s as well as oligomeric and polymeric derivatives of benzophenone-, thioxanthone-, and aminobenzoate-type have been developed for food packaging applications. PI 6 is a 408 amu monomeric PI with low volatility and low migratory tendencies. However, as observed in this experiment, PI 6 readily decomposes when exposed to UV-energy to give off 2-hydroxy-2-methyl-1-phenylpropanone (PI 1), another PI molecule which is low molecular weight and highly migratory. Polymeric PI’s have molecular weights in excess of 1000 amu and pose little risk for migration. Despite this, decomposition of these compounds under UV-light may occur, producing volatile and migratory low molecular weight molecules. These polymeric PI’s have found little use in food packaging systems to date due to increased cost, higher viscosity, and lower photoinitiation efficiency compared to low molecular weight PI’s [25].

Food safety regulatory authorities throughout the world currently consider migration of PI’s and their photolytic decomposition products in different ways. In the United States, FDA has a general provision for impurities of indirect food additives under 21 CFR 174.5. In this article, FDA recommends submission of the major impurities of food additives (e.g., residual starting materials, byproducts, degradation products) when filing food contact notifications or food additive petitions for food contact substances [26]. Specific impurities may be exempted under 21 CFR 170.39 if their dietary concentration is below the threshold of regulation. The threshold of regulation under this guideline for compounds migrating from food-contact articles is a dietary concentration of 0.5 ppb. This concentration corresponds to 1.5 µg of a compound per person per day, assuming a daily intake of 1500 g of solid food and 1500 g of liquid food per day. Validated migration testing data with dietary exposure calculations are required to demonstrate conformity with these regulations [27]. Compounds detected below the 0.5 ppb dietary concentration threshold are considered exempt from regulation unless evidence exists that the compound may be carcinogenic or may present other health or safety concerns. When considering dietary exposure calculations for food packaging migrants, consumption factors for different food packaging substrates must be taken into account. The consumption factor refers to the fraction of the daily diet expected to contact specific packaging materials [22]. Another reason for FDA regulatory exemption is the listing of a compound as generally recognized as safe (GRAS) for a specific application, such as a food ingredient or packaging component. For a compound to become GRAS listed, the petitioner must submit extensive chemical and toxicological data demonstrating the safety of the compound in its intended application. GRAS petitions are reviewed by FDA and may approve compounds for use only in specified applications and at clearly-defined usage levels [12]. Benzoic acid and 4-aminobenzoic acid are GRAS-listed compounds identified as photolytic decomposition products of PI’s [28].

The list of GRAS compounds is a living document and changes can be made to it when data deems it appropriate. For example, benzophenone, a commonly used PI, flavor compound, and rubber plasticizer, had its GRAS status revoked in 2018, effective October 2020, based on studies showing carcinogenicity in rats [29]. Benzophenone will no longer be permitted for use as a flavorant or as a food packaging additive, and food packaging and coating manufacturers must reformulate product lines to replace benzophenone as a PI. California Proposition 65, the Safe Drinking Water and Toxic Enforcement Act of 1986, was enacted to inform citizens of potential exposure to chemicals demonstrated to cause cancer, birth defects, or reproductive harm [11]. Benzophenone was added to the Proposition 65 list in 2012 based on evidence of carcinogenicity.

The European Food Safety Authority (EFSA) set forth guidelines for food contact materials and migration testing in guidance document EC 1935/2004, which states that migratory chemical compounds from food packaging materials may not endanger human health, cause an unacceptable change in the characteristics of the food product, or deteriorate the organoleptic properties of the food [30]. Guidance document EC 10/2011 provides regulatory guidelines including migration test protocols and allowable levels of migration for plastic food contact articles. This document sets forth specific migration limits (SML) for specific chemical compounds based on toxicological data. The SML for benzophenone is 0.6 mg/kg of food product (60 ppb *w*/*w*). For compounds with no specified SML, a general concentration limit of 0.1 mg/kg of food product (10 ppb *w*/*w*) is specified. EC 10/2011 defines non-intentionally added substances (NIAS) as impurities, reaction intermediates, and decomposition products of components of food contact materials, and photolytic decomposition products of PI’s fall under this category [13]. None of the photolytic decomposition products identified in the present study have specified SML, so the general limit of 10 ppb *w*/*w* applies to these compounds.

Other food safety regulatory authorities have issued guidance documents regarding food additives and food packaging migration. Several of these documents, such as the Swiss Ordinance issued by the Swiss Confederation, include regulations for allowable PI’s with SML for food contact material applications, but photolytic decomposition products of PI’s are not discussed [10].

## 4. Materials and Methods

### 4.1. Food Packaging Samples

In this study, 258 different food packaging samples each containing UV-printed inks, coatings, or adhesives were analyzed. Sources of packaging samples included commercial, experimental, and developmental food packaging materials. Developmental materials were provided for migration testing in support of regulatory petitions. Samples were submitted by over 100 food packaging manufacturers for routine migration testing. Sample substrates included paperboard, polyethylene films, polyethylene terephthalate films, polystyrene, and various laminate structures. Samples were provided in various forms, including reeled films, stacked paperboards, formed pouches, and nested cups or containers. Detailed descriptions of individual samples are beyond the scope of this publication. The UV-curable formulations used in the samples were varied and represent the broad range of PI’s commonly used in food packaging materials.

### 4.2. Reagents, Reference Standards, and Materials

Methylene chloride (Optima Grade) and acetonitrile (HPLC Grade) were obtained from Fisher Scientific (San Jose, CA, USA). Ethanol (190 proof, USP Grade) was obtained from Pharmco Products (Brookfield, CT, USA). Formic acid (Reagent Grade) and anthracene-d10 (CAS No. 1719-06-8, 98% isotopic purity) were obtained from Sigma-Aldrich (Milwaukee, WI, USA). Distilled and deionized water was prepared in-house by double distillation in a glass-lined still followed by deionization and activated carbon filtration in a Millipore Milli-Q purification system.

### 4.3. UV-Irradiation of PI Standards

Sample supports were prepared by wrapping a rigid paperboard disk with pre-cleaned aluminum foil. Supports were used to provide a rigid, easy to handle medium for UV-irradiation of PI standards. PI standards were dissolved in methylene chloride at a concentration of 10 mg/mL. One milliliter of solution was pipetted onto a foil support, and the solvent was evaporated using mild heat, leaving 10 mg of the PI on the foil. The foil supports with dried sample were stored in individual petri dishes, placed in a desiccator, and protected from light until the UV-exposure process.

PI standards were UV-irradiated using a UVEXS LCU750D (UVEXS Incorporated, Sunnyvale, CA, USA) pilot scale UV-curing apparatus. The system was equipped with a 300 watt/inch medium pressure mercury lamp, which emits UV-energy across 200–600 nm. The sample conveyor belt was run at 25% of its maximum speed. Foil supports containing deposited PI residue were placed on the conveyor belt and passed under the UV lamp once each, then returned to petri dishes and stored away from light. Following UV-irradiation, the foil was carefully removed from the paperboard support, and the PI residue was dissolved from the foil with 5–10 mL of methylene chloride. The foil was rinsed several times to ensure full dissolution of the residue. The methylene chloride extracts were transferred to 5 mL glass conical-bottom Reacti-Vials (Thermo Fisher Scientific, San Jose, CA, USA) and concentrated to 1.0 mL under a gentle stream of nitrogen at room temperature for GC-MS analysis. Concentrated extracts were then analyzed by GC-MS under the conditions described in the following section.

### 4.4. Migration Testing

Sections of packaging material were cut and placed into the migration cells with the food-contact side (unprinted side) exposed to the food simulant. Food simulants used were 10% ethanol and 95% ethanol, corresponding to recommended simulants in FDA guidance. For packaging materials tested in extraction cells, the food simulant volume to surface area ratio was 1.57 mL/cm^2^ (10 mL/inch^2^), also corresponding to FDA guidance. For materials tested in the container, actual volume to surface area ratio was used. Extraction cells or containers were incubated corresponding to FDA Condition of Use A-H, using appropriate conditions for the intended use of the packaging material. Method blanks consisting of empty extraction cells filled with food simulants were routinely prepared, and compounds detected in the method blanks were disregard in the data treatment of the test samples. Comparable migration tests by our group have been previously described [14].

### 4.5. Sample Preparation of Migration Test Extracts for GC-MS Analysis

Following incubation, extraction cells were drained into borosilicate glass test tubes sealed with Teflon-lined screw cap closures and stored away from light prior to analysis. In preparation for GC-MS analysis, back extraction of the extracts into methylene chloride was used to minimize evaporative loss during the concentration step. PI’s and their photolytic decomposition products are mostly hydrophobic and are preferentially soluble in methylene chloride rather than aqueous solvents. Aliquots of 10% ethanol extracts (40 mL) were transferred to borosilicate glass test tubes with Teflon-lined closures and matrix-spiked with 100 parts per billion *w/v* (ppb *w*/*v*) anthracene-d10 to serve as an internal standard via the addition of 4.0 µL of 1.0 mg/mL stock solution in methylene chloride. 5.0 mL of methylene chloride were added, and the tubes were stoppered tightly, vigorously extracted, and then centrifuged at 2500 rpm for 30 min to promote complete phase separation. After centrifugation, the lower methylene chloride layer was removed using a Pasteur pipet, transferred to 5.0 mL borosilicate glass Reacti-Vials and concentrated to 0.1–0.5 mL under a gentle stream of nitrogen at room temperature without reaching dryness. For 95% ethanol extracts, the same procedure was followed, but extracts were diluted with distilled water to 10% ethanol prior to methylene chloride extraction. The ratio of 10% ethanol to methylene chloride was maintained at a constant 8:1 to avoid altering aqueous-organic partitioning coefficients. Concentrated extracts were then analyzed by GC-MS under the conditions described in the following section.

### 4.6. GC-MS Analysis Methodology

Two different GC-MS instruments were used in this study because the packaging migration data were accumulated over a period of 10 years in our laboratories. Despite this, the EI mass spectra acquired were equivalent regardless of the instrument upon which they were acquired.

GC-MS analyses of UV-irradiated PI samples were conducted using a Finnigan Trace-GC Ultra 8000 gas chromatograph (Thermo Fisher Scientific, San Jose, CA, USA) interfaced to a Fisons MD800 single stage quadrupole mass spectrometer (Waters Corporation, Milford, MA, USA) equipped with an X-calibur data system. The GC was equipped with a 30 m × 0.22 mm I.D. Equity 5 (Supelco, Bellefonte, PA, USA) (5% phenyl, poly-dimethylphenylsiloxane) capillary column with a 0.25 µm film thickness. Analytical method blanks were analyzed between PI samples in order to account for carryover between samples. Packaging migration samples were analyzed using a Varian 3400 GC interfaced to a Thermo TSQ7000 triple stage quadrupole mass spectrometer (Thermo Fisher Scientific, San Jose, CA, USA) equipped with an X-calibur data system. This GC was equipped with a 30 m × 0.32 mm I.D. Guardian ZB-5MS (Phenomenex, Torrance, CA, USA) capillary column with a 0.25 µm film thickness.

GC and MS conditions were the same for both instruments. Injection volume was 1.0 µL. The injector was maintained at 300 °C in splitless mode with a 100:1 split activated 0.5 min after injection to serve as a septum purge. The GC column was temperature programmed from 50 °C (hold 3 min) to 320 °C at a rate of 10 °C/min, holding at the upper limit for 10 min. The GC-MS transfer line was maintained at 320 °C. The mass spectrometers were operated in EI mode (70 eV) scanning 35–800 *m*/*z* once each second. For data management, total ion current (TIC) chromatograms were integrated within the data system software and the peak lists and area integration values were pasted into spreadsheet-based templates. For migration analyses, semi-quantitative concentration values for migratory compounds were calculated based on TIC peak area relative to matrix-spiked anthracene-d10 internal standard, assuming a detector response factor of 1.0. Analytical method blanks were analyzed between packaging extracts in order to account for carryover between samples.

### 4.7. ESI-MS Analysis Methodology

UV-irradiated PI samples were also analyzed by electrospray ionization (ESI) mass spectrometry in order to observe nonvolatile dimers and higher oligomers of PI’s formed during UV-irradiation. Methylene chloride extracts of irradiated PI’s were evaporated to dryness under a gentle stream of nitrogen at room temperature and re-dissolved in 1.0 mL acetonitrile containing 0.1% formic acid. Loop injections (5 µL) were used to infuse sample extracts into ESI ion source of a Thermo LCQ ion trap mass spectrometer (Thermo Fisher Scientific, San Jose, CA, USA). Mobile phase flow consisting of acetonitrile containing 0.1% formic acid was delivered at 150 µL/min by a Finnigan Surveyor MS-grade quaternary pump (Thermo Fisher Scientific, San Jose, CA, USA). The mass spectrometer was operated in both positive and negative ion modes scanning 100–1500 *m*/*z*. Sheath gas flow was set at 60 arbitrary units. Auxiliary gas flow was set at 0 arbitrary units. Spray voltage was set at 5.1 kV. Capillary temperature was maintained at 220 °C, and capillary voltage was set at 39 V.

## 5. Conclusions

In this study, 113 photolytic decomposition products were identified from 24 PI’s commonly used in food packaging inks, coatings, and adhesives. The data presented for these photolytic decomposition products including their parent PI’s, Kovats retention index, migratory tendencies, and other identification information can serve as important references to assist researchers in the tracing of sources of photolytic decomposition products and other non-intentionally added substances when they are found in food packaging material analyses.

Based on our migration testing on 258 UV-cure food packaging samples, 18 PI’s and 21 photolytic decomposition products were found to have migrated through the substrates. The most commonly observed migrants were 2,4,6-trimethylbenzaldehyde, 1-phenyl-2-butanone, 4-methylthiobenzaldehyde, and benzaldehyde. Many of these photolytic decomposition products were observed to migrate at high concentrations, and many of them have not been evaluated for toxicological safety upon consumption or contact. While regulatory emphasis is increasing on non-intentionally added substance migrants, the present study will be of great value for future risk assessment for UV-cure food packaging systems and act as an important database for food packaging manufacturers to maintain systematic quality control of their packaging products.

## Figures and Tables

**Figure 1 molecules-24-03592-f001:**
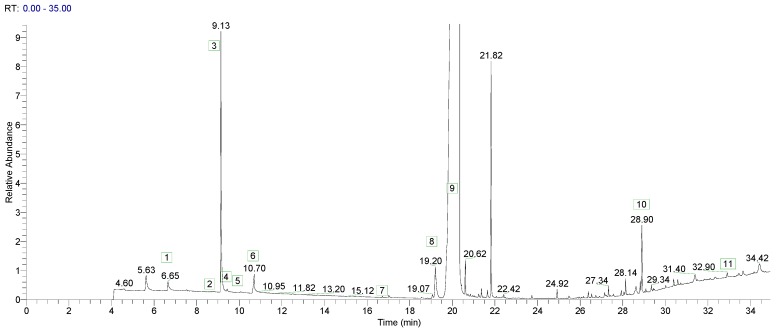
GC-MS total ion current chromatogram of UV-exposed PI 10. Photolytic decomposition product peaks are numbers as follows: 1. benzaldehyde, 2. acetophenone, 3. methyl benzoate, 4. benzaldehyde dimethyl acetal, 5. methyl phenyl carbonate, 6. benzoic acid, 7. diphenyl ether, 8. diphenylethanedione, 9. 2,2-dimethoxy-2-phenylacetophenone (parent PI), 10. dimer of parent PI, 11. trimer of parent PI.

**Figure 2 molecules-24-03592-f002:**
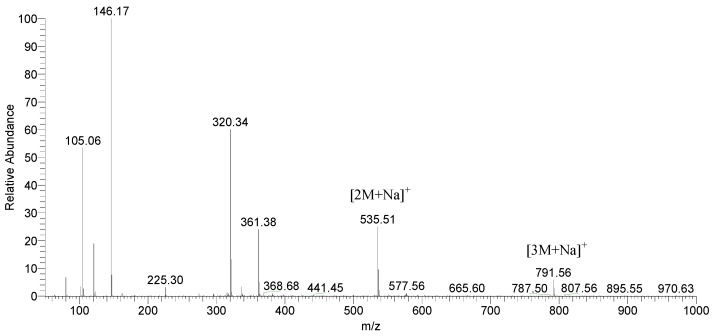
ESI-MS mass spectrum of UV-exposed PI 10. The dimer and trimer adduct of 10 are seen as sodiated adducts ([2M + Na]^+^ and [3M + Na]^+^) at *m*/*z* 535.51 and 791.56, respectively.

**Table 1 molecules-24-03592-t001:** Photoinitiator Molecules Analyzed.

Reference Number	CAS Number	IUPAC Nomenclature	Structure
1	7473-98-5	2-hydroxy-2-methyl-1-phenylpropanone	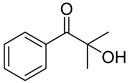
2	947-19-3	1-hydroxycyclohexyl-1-phenylmethanone	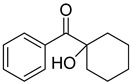
3	106797-53-9	1-[4-(2-hydroxyethoxyl)-phenyl]-2-hydroxy-2-methyl propanone	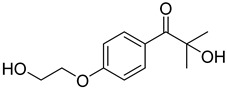
4	163702-01-1	oligo{2-hydroxy-2-methyl-1-[4-(1-methylvinyl)phenyl] propanone}	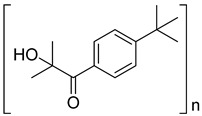
5	474510-57-1	2-hydroxy-1-{4-[4-(2-hydroxy-2-methylpropionyl)benzyl] phenyl)-2-methylpropan-1-one	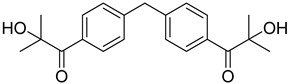
6	163702-01-0	2-hydroxy-1-{1-[4-(2-hydroxy-2-methylpropionyl) phenyl]1,3,3-trimethylindan-5-yl-}2-methylpropan-1-one	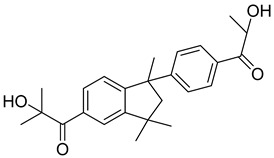
7	71868-10-5	2-methyl-1-[4-(methylthio) phenyl]-2-morpholinopropan-1-one	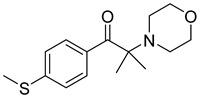
8	119313-12-1	2-benzyl-2-(dimethylamino)-4’-morpholinobutyrophenone	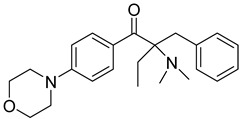
9	119344-86-4	2-dimethylamino-2-(4-methylbenzyl)-1-(4-morpholin-4-yl-phenyl) butan-1-one	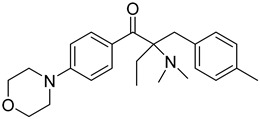
10	24650-42-8	2,2-dimethoxy-2-phenylacetophenone	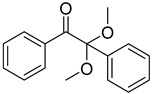
11	75980-60-8	2,4,6-trimethylbenzoyl-diphenylphosphine oxide	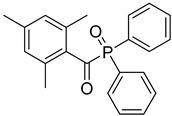
12	84434-11-7	ethyl 2,4,6-trimethylbenzoylphenyl phosphinate	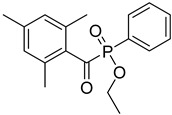
13	162881-26-7	bis(2,4,6-trimethylbenoyl)phenylphosphine oxide	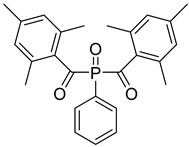
14	119-61-9	benzophenone	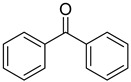
15	134-84-9	4-methylbenzophenone	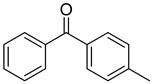
16	954-16-5	2,4,6-trimethylbenzophenone	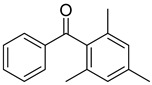
17	2128-93-0	4-phenylbenzophenone	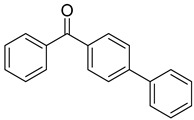
18	83846-85-9	4-(4-methylphenylthio) benzophenone	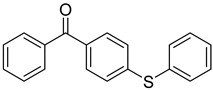
19	606-28-0	methyl 2-benzoylbenzoate	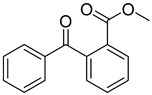
20	83846-86-0	2-isopropylthioxanthone	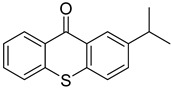
21	82799-44-8	2,4-diethylthioxanthone	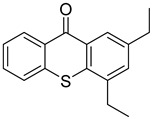
22	90-93-7	4,4’-bis(diethylamino) benzophenone	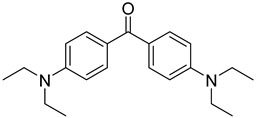
23	21245-02-3	2-ethylhexyl 4-dimethylaminobenzoate	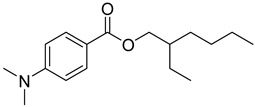
24	10287-53-3	ethyl 4-dimethylaminobenzoate	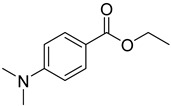

**Table 2 molecules-24-03592-t002:** Photolytic Decomposition Products Identified.

Photolytic Decomposition Product	CAS Number	MW	Parent PI	Peak Area % ^1^	RI ^2^
*N*-methylpropanamide	1187-58-2	87	8	0.019	1026
			9	0.010	
cyclohexanone	108-94-1	98	2	0.458	898
cyclohexanol	108-93-0	100	2	0.004	894
*N*,*N*-dimethylpropanamide	758-96-3	101	8	0.056	973
			9	0.029	
benzaldehyde	100-52-7	106	1	0.186	966
			2	0.264	
			4	0.006	
			8	0.033	
			10	0.105	
*p*-xylene	106-42-3	106	9	0.159	872
phenylphosphine	638-21-1	110	13	0.069	917
2-hydroxycyclohexanone	533-60-8	114	2	0.069	1003
2-methoxythiazole	14542-13-3	115	7	0.063	1188
*N*,*N*-dimethyl-3-methoxypropylamine	20650-07-1	117	9	0.023	1592
acetophenone	98-86-2	120	1	0.123	1080
1,3,5-trimethylbenzene	108-67-8	120	11	0.319	971
			12	4.832	
			13	4.526	
			15, 16 ^6^	0.002	
1-phenylethenol ^4^	4383-15-7	120	3	0.013	1112
*N*,*N*-dimethylaniline	121-69-7	121	23	0.002	1098
benzoic acid	65-85-0	122	1	0.732	1126
			2	0.107	
			10	0.269	
			10	0.012	
4-hydroxybenzaldehyde	123-08-0	122	3	0.027	1674
4-methylbenzenethiol	106-45-6	124	18	0.002	1075
methyl phenyl sulfide	100-68-5	124	7	0.014	1092
4-acetylmorpholine	1696-20-4	129	7	0.038	1259
(E)-2-phenyl-2-butene	768-00-3	132	8	0.032	1063
(E)-1-phenyl-1-butene	1005-64-7	132	8	0.007	1132
1,3-diethylbenzene	141-93-5	134	21	0.003	1054
benzothiazole	95-16-9	135	7	0.003	1243
methyl benzoate	93-58-3	136	1	0.013	1103
			10	1.248	
2,4,6-trimethylphenol	527-60-6	136	11	0.015	1216
			12	0.399	
			13	0.128	
			15, 16 ^6^	0.002	
α,α-dimethylbenzyl alcohol	617-94-7	136	1	0.249	1282
4-aminobenzoic acid	150-13-0	137	24	0.001	1594
4-methoxybenzenethiol	696-63-9	140	7	0.027	1405
2-methyl-1-phenyl-1-butene	56253-64-6	146	9	0.034	1170
3-phenyl-2-propenamide	621-79-4	147	9	0.001	1178
6-methyl-1,2,3,4-tetrahydroquinoline	91-61-2	147	8	0.008	1382
benzoyl methyl ketone	579-07-7	148	1	0.021	1178
1-phenyl-2-butanone	1007-32-5	148	8	1.050	1230
4-isopropylbenzaldehyde	123-03-2	148	4	0.001	1252
2,4,6-trimethylbenzaldehyde	487-68-3	148	11	0.113	1314
			12	10.05	
			13	0.996	
*N*,*N*-diethylaniline	91-66-7	149	22	0.004	1232
*N*,*N*-dimethylphenethylamine	1126-71-2	149	8	0.036	1500
methyl phenyl carbonate	13509-27-8	152	10	0.024	1155
benzaldehyde dimethyl acetal	1122-88-8	152	10	0.039	1118
4-(methylthio)benzaldehyde	3446-89-7	152	7	1.440	1442
biphenyl	92-52-4	154	14	0.010	1393
			17	0.006	
3-methyl-1-phenyl-3-buten-2-one	55956-30-4	160	8	0.004	1318
1-phenyl-1-penten-3-one	3152-68-9	160	8	0.011	1322
4′-isopropenylacetophenone	5359-04-6	160	4	0.005	1334
2-methyl-1-phenyl-1-butanone	938-87-4	162	8	0.010	1286
2-hydroxy-2-methylpropiophenone	7473-98-5	162	6	0.057	1295
4-phenylmorpholine	92-53-5	163	8	0.094	1439
			9	0.146	
*N*,*N*-dimethylbenzeneacetamide	18925-69-4	163	8	0.019	1498
2,4,6-trimethylbenzoic acid	480-63-7	164	11	5.351	1467
			12	1.390	
			13	44.52	
2-(2,4,6-trimethylphenyl)ethanol ^4^	6950-92-1	164	13	0.029	1246
2-ethoxy-1,3,5-trimethylbenzene ^4^	61248-63-3	164	12	0.176	1249
4-(dimethylamino)benzoic acid	619-84-1	165	23	0.002	2131
4-(methylthio)acetophenone	1778-09-2	166	7	0.025	1547
3-(4-methoxyphenyl)-1-propanol	5406-18-8	166	3	0.567	1604
3*H*-1,2-benzodithiol-3-one	1677-27-6	167	20	0.009	1568
4-(methylthio)benzoic acid	13205-48-9	168	7	0.353	1620
diphenylmethane	101-81-5	168	14	0.011	1448
diphenyl ether	101-84-8	170	2	0.005	1411
			10	0.001	
			14	0.005	
			15	0.001	
			15, 16 ^6^	0.001	
ethyl phenylphosphinate ^4^	2511-09-3	170	12	64.82	1418
cyclopentyl phenyl ketone	5422-88-8	174	2	0.016	1574
2-methyl-1-phenyl-1,3-butanedione	6668-24-2	176	9	0.001	1340
methyl 2,4,6-trimethylbenzoate	2282-84-0	178	11	0.001	1355
			12	0.868	
			13	0.010	
4-morpholinylaniline	2524-67-6	178	9	0.001	2032
methyl 4-(dimethylamino)benzoate	1202-25-1	179	23	0.001	1651
1-methoxyethyl benzoate	51835-44-0	180	15	0.002	1295
triethyl phosphate	78-40-0	182	12	0.271	1160
benzophenone	119-61-9	182	18	0.001	1644
2,2′-dimethylbiphenyl	605-39-0	182	15, 16 ^6^	0.004	1524
1,2-diphenylethane	103-29-7	182	8	0.498	1539
2-(3-methylphenoxy)pyridine ^4^	1793003-62-9	185	9	0.001	1694
diphenylphosphine	829-85-6	186	12	0.035	1585
2-phenoxyphenol	2417-10-9	186	2	0.010	1681
cyclohexyl phenyl ketone	712-50-5	188	2	0.006	1620
4-morpholinobenzaldehyde	1204-86-0	191	8	1.093	1910
ethyl 2,4,6-trimethylbenzoate	1754-55-8	192	11	0.018	1829
4-(diethylamino)benzoic acid	5429-28-7	193	22	0.006	1854
			15, 16 ^6^	0.009	
1-(methylthio)-4-(2-methyl-3-hydroxy-2-propene)-yl-benzene ^4^	N/A	194	7	0.350	1675
2,2′-dimethyldiphenylmethane ^4^	1634-74-8	196	15, 16 ^6^	0.007	1632
p-morpholinoacetophenone	39910-98-0	205	9	0.020	1990
methyl 4-(diethylamino) benzoate ^4^	91563-80-3	207	22	0.008	1833
4′-(2-hydroxyethoxy)-2-methylpropiophenone ^4^	N/A	208	3	0.692	1858
diphenylethanedione	134-81-6	210	1	0.488	1857
			2	0.121	
			10	0.297	
1,2-di(4-methylphenyl)ethane	538-39-6	210	9	0.897	1800
thioxanthone	492-22-8	212	20	0.010	2129
diethyl phenylphosphonate	1754-49-0	214	12	16.59	1514
methyldiphenylphosphine oxide	2129-89-7	216	11	0.019	2028
diphenylphosphinic acid	1707-03-5	218	13	0.007	1897
4-hydroxychalcone	2657-25-2	224	20	0.001	1857
methyl diphenylphosphinate	1706-90-7	232	12	0.122	1951
2-ethylhexyl benzoate	5444-75-7	234	23	0.001	1746
diphenylpropanetrione	643-75-4	238	15, 16 ^6^	0.365	2041
3-methyl-1,8,9-anthracenetriol	491-59-8	240	21	0.028	2361
bis(4-methylphenyl)disulfide	103-19-5	246	18	0.102	2082
ethyl diphenylphosphinate^4^	1733-55-7	246	11	0.200	1988
			12	0.445	
2,2-dimethyl-1,3-diphenyl-1,3-propanedione	41169-42-0	252	4	0.001	2021
2-ethylhexyl-2-ethylhexanoate	7425-14-1	256	23	0.008	1608
*N*,*N*’-diethyl-N,N’-diphenylurea	85-98-3	268	22	0.006	1914
dimer of 2,4,6-trimethylphenol ^3^	N/A	272	13	0.039	2192
*N*-hydroxy-2-methoxy-*N*-methylbenzene Carbothioamide ^4^	95096-17-6	273	7	0.111	1878
*N*-benzyl-*N*-hydroxy-2-methoxy benzenecarbothioamide	93979-07-8	273	7	0.010	2481
*N*-ethyl-2-(ethylphenylamino)-*N*-phenylacetamide ^4^	N/A	282	22	0.002	1978
2,4,6-trimethylphenyl benzoate ^4^	N/A	282	15, 16 ^6^	0.007	1632
*O*-dibenzoylbenzene	1159-86-0	286	19	0.014	2516
dimer of 2,4,6-trimethylbenzaldehyde ^3^	N/A	296	13	5.168	2063
4-methyl-2-(1-phenyl ethyl)phenyl benzoate	18062-71-0	316	1	0.077	2617
dimer of 2,4,6-trimethylacetophenone ^3^	N/A	324	13	0.152	2258
dimer of PI 1 ^3^	N/A	328	1	1.531	2219
1-[(4-diethylamino)phenyl]-1-[(4-diethylamino-3-methylamino)phenyl]methanone ^4^	N/A	353	22	0.026	3231
1,1,2,2-tetraphenyl-1,2-ethanediol	464-72-2	366	14	0.012	2189
dimer of PI 4 ^5^	163702-01-0	408	4	98.39	3020
dimer of PI 10 ^3^	N/A	512	10	0.395	2941
dimer of PI 11 ^3^	N/A	696	11	8.898	3212
trimer of PI 10 ^3^	N/A	768	10	0.036	3272

^1^ Area percent calculated from GC-MS total ion current integration.^2^ Kovats retention index on DB-5 (5% phenyl, 95% methyl capillary column) calculated from n-alkanes.^3^ Exact structure not determined.^4^ Tentative structural identification by mass spectral interpretation.^5^ Identified by both GC-MS and ESI-MS. ^6^ Compound identified from commercially-available mixture of 15 and 16.

**Table 3 molecules-24-03592-t003:** Frequency of Occurrence and Concentration Range of PI’s and Photolytic Decomposition Products.

Name	CAS Number	Frequency of Occurrence ^1^	10% Ethanol ^2^ Concentration Range (ng/cm^2^)	95% Ethanol ^3^ Concentration Range (ng/cm^2^)
benzophenone (PI 14)	119-61-9	88	0.24–62.84 ^c^	1.38–66.30 ^c^
			0.04–948.37 ^e^	0.15–266.20 ^e^
				6.61–9.64 ^m^
2-hydroxy-2-methyl-1-phenylpropanone (PI 1)	7473-98-5	139	0.6 ^a^	1.22–17.98 ^a^
			0.01–499.55 ^c^	19.42–737.87 ^c^
			0.25–361.17 ^e^	0.53–1556.64 ^e^
			65.65 ^h^	81.08 ^h^
1-hydroxycyclohexyl-1-phenylmethanone (PI 2)	947-19-3	73	1426.48^c^	49.09–2238.76 ^c^
			0.27–1169.02 ^e^	1.55–1163.71 ^e^
			753.25 ^h^	1292.84 ^h^
2,2-dimethoxy-2-phenylacetophenone (PI 10)	24650-42-8	49	9.27–673.67 ^c^	14.55–2141.64 ^c^
			0.05–109.75 ^e^	0.55–454.63 ^e^
			80.22 ^h^	344.30 ^h^
			0.33–0.99 ^m^	0.85–3.97 ^m^
methyl 2-benzoylbenzoate (PI 19)	606-28-0	56	4.29–1854.87^c^	3376.41 ^c^
			0.10–786.09^e^	3.33–1072.49 ^e^
			520.18 ^h^	471.65 ^h^
2-methyl-1-[4-(methylthio) phenyl]-2-morpholinopropan-1-one (PI 7)	71868-10-5	83	3.89–2090.98 ^c^	2.95–4429.78 ^c^
			0.28–365.98 ^e^	0.93–1209.68 ^e^
			156.68 ^h^	671.34 ^h^
2-benzyl-2-(dimethylamino)-4′-morpholinobutyrophenone (PI 8)	119313-12-1	3	6.79 ^e^	39.02–51.98 ^e^
2,4,6-trimethylbenzoyl diphenylphosphine oxide (PI 11)	75980-60-8	21	236.27–702.12 ^c^	40.64–1603.21 ^c^
			6.40–121.20 ^e^	79.17–9122.66 ^e^
ethyl 2,4,6-trimethylbenzoyl phenylphosphinate (PI 12)	84434-11-7	2		435.63–904.32 ^e^
4-phenylbenzophenone (PI 13)	2128-93-0	16	14.12–16.73 ^c^	6.29–155.79^c^
			1.83–51.71 ^e^	0.73–13.07^e^
			26.27 ^h^	41.37^h^
4-methylbenzophenone (PI 15)	134-84-9	7		11.93–87.54 ^e^
2-isopropylthioxanthone (PI 20)	83846-86-0	20	0.32 ^c^	7.97 ^c^
			1.88–178.97 ^e^	4.68–3221.92 ^e^
			166.04 ^h^	1550.84 ^h^
2,4-diethylthioxanthone (PI 21)	82799-44-8	25	11.21–11.34 ^c^	896.93–2762.95 ^c^
			1.03–2.45 ^e^	5.27–2227.32 ^e^
ethyl 4-dimethyl aminobenzoate (PI 24)	10287-53-3	70	0.29–892.28 ^c^	22.647–1637.24 ^c^
			0.37–439.49 ^e^	0.50–1701.19 ^e^
			153.09–1746.04 ^h^	255.19–389.86 ^h^
2-ethylhexyl- 4-dimethylaminobenzoate (PI 23)	21245-02-3	6		59.62–1019.10 ^e^
4,4′-bis(diethylamino) benzophenone (PI 22)	90-93-7	6	0.86–11.48 ^e^	0.99–3832.71 ^e^
{2-hydroxy-2-methyl-1-[4-(1-methylvinyl)phenyl] propanone} (PI 4)	163702-01-1	2	11.99^a^	6.96 ^a^
benzaldehyde	100-52-7	78	0.17–33.06 ^c^	1.61–62.01 ^c^
			0.03–111.20 ^e^	0.44–68.86 ^e^
			11.20 ^h^	12.96 ^h^
acetophenone	98-86-2	38	16.49 ^c^	29.13 ^c^
			0.22–10.95 ^e^	0.24–24.76 ^e^
			3.75 ^h^	2.23 ^h^
methyl benzoate	93-58-3	33	0.19–450.52 ^c^	1.61–548.03 ^c^
			0.10–36.05 ^e^	0.18–78.87 ^e^
			2.11 ^h^	8.59 ^h^
				1.47–2.89 ^m^
2,4,6-trimethylbenzaldehyde	487-68-3	130		24.57 ^a^
			0.03–1938.56 ^c^	0.84–26.96 ^c^
			0.24–22.22 ^e^	0.26–64.59 ^e^
			8.00^h^	8.12 ^h^
4-methylthiobenzaldehyde	3446-89-7	81	0.22–177.49^c^	0.69–267.70 ^c^
			0.40–24.80^e^	0.39–237.19 ^e^
			28.39^h^	
1-phenyl-2-butanone	1007-32-5	83	1.14^a^	2.70–50.90 ^a^
			0.07–0.81^c^	10.55–12.69 ^c^
			0.06–440.83 ^e^	0.60–423.52 ^e^
			0.92–1.87 ^g^	
			6.33–13.69 ^h^	0.31–4.70 ^h^
α,α-dimethylbenzyl alcohol	617-94-7	68	7.87–25.09 ^c^	0.63–23.44 ^c^
			0.05–18.68 ^e^	0.28–258.54 ^e^
			1.44 ^h^	1.74 ^h^
2,4,6-trimethylbenzoic acid	480-63-7	44	17.14–52.12 ^c^	7.32–92.44 ^c^
			0.10–14.38 ^e^	0.35–156.39 ^e^
			0.29 ^h^	
benzothiazole	95-16-9	12		1.29 ^c^
			0.18–5.41 ^e^	0.19–1.32 ^e^
biphenyl	92-52-4	6	0.20–4.16 ^e^	12.57 ^e^
cyclohexanone	108-94-1	15	1.51–52.71 ^e^	2.34–27.56 ^e^
p-diisopropenylbenzene	1605-18-1	26	0.62 ^a^	1.99–31.76 ^a^
				62.00d
			0.05–3.60 ^e^	0.51–26.58 ^e^
diphenylethanedione	134-81-6	6	1.50–7.34 ^e^	8.06–153.78 ^m^
ethyl 2,4,6-trimethylbenzoate	1754-55-8	4		5.53–44.54 ^e^
diethyl phenylphosphonate	1754-49-0	4		5.16–24.09 ^e^
1,3,5-trimethylbenzene	108-67-8	4		14.55–63.48 ^c^
2,4,6-trimethylphenol	527-60-6	7		0.33–83.37 ^e^
			1.35 ^h^	1.93 ^h^
benzoic acid	65-85-0	20	38.58–67.23 ^c^	1.69–27.09 ^c^
				0.77–10.79 ^e^
2-hydroxy-2-methyl-1-phenylpropanone	7473-98-5	1	17.91 ^e^	
methyl-4-dimethyl aminobenzoate	1202-25-1	7	0.56 ^c^	1.14–2.01 ^c^
			5.86–14.49 ^e^	26.26–44.77 ^e^
4-diethylaminobenzoic acid	5429-28-7	1		25.81 ^e^

^1^ Absolute frequency of occurrence in 258 total food-contact side migration samples examined. ^2^ 10% ethanol food simulant used for samples intended for aqueous and acidic food types. ^3^ 95% ethanol food simulant used for samples intended for fatty food types. ^a^ FDA condition of use “A”; 2 h at 121 °C + 10 days at 40 °C; high temperature heat sterilized. ^c^ FDA condition of use “C”; 66 °C for 2 h + 10 days at 40 °C; hot filled or pasteurized above 66 °C. ^e^ FDA condition of use “E”; 10 days at 40 °C; room temperature filled and stored. ^f^ FDA condition of use “F”; 10 days at 20 °C; refrigerated storage. ^g^ FDA condition of use “G”; 5 days at 20 °C; frozen storage. ^h^ FDA condition of use “H”; 2 h at 100 °C; frozen storage, intended to be reheated in container. ^m^ FDA microwave condition; 15 min at 100 °C for 10% ethanol, 15 min at 130 °C for 95% ethanol.

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
