# Peer review of "Identification and Migration Studies of Photolytic Decomposition Products of UV-Photoinitiators in Food Packaging"

_molecules, 2019, doi:10.3390/molecules24193592_

Round 1
Reviewer 1 Report
Comments:
The current manuscript deals with the identification of decomposition products of UV-photoinitiators in various food packaging samples. These photoinitiators use for the polymerisation process or inks, coatings, …. The problem and the functionality of the UV-photoinitiators are well-known. However, the design of experiment is good and the amount of samples is high in this study. Therefore, new knowledge is gained by this study and the publishing of this manuscript can be supported. However, below the authors will find some comments for improvements:
Abstract:
The conclusion part in the abstract is missing. What can you do with these results, what are the next steps?
IntroductionPlease change your stile to describe the current situation of the state of the art (e.g., this review is incomplete, or this study is also incomplete). It is not enough to say that the other studies were incomplete.
ResultsThe quality of Figure 1 and Figure 2 is not very good. The authors should think if all both Figures are really necessary. What is the additional value of the Figures?
Materials and Methods4.3. UV-Irradiation of PI Standards
Please mention also the wavelength range of the UV lamp (or energy provided by the UV-light) of the UV-irradiation
References
Please change the references according to the guidelines of the journal (e.g., use correct journal abbreviations)
Reviewer 2 Report
The manuscript by Scarsella, Zhang, and Hartman describes a very interesting study on the migration and investigation of Photolytic Decomposition Products of UV-Photoinitiators in Food Packaging. The aim of the paper is to isolate and identify the photolytic decomposition products and, additionally, to measure the frequencies of occurrence of PI’s and their photolytic decomposition products in food packaging material migration data.
The paper is very interesting and deserve publication in Molecules. I only have a couple of minor suggestions:
1) The paper is well written but some sections are, in my opinion, too dense of information. The Discussion section could be better schematized
2) Thea authors report that: aims to isolate and identify the photolytic decomposition products formed from PI’s ”under typically used industrial conditions” but they also specify that In the present ”experiments, however, no reactant monomer was provided”.
Please highlight if these differences and/or UV radiation exposure (time, energy. etc.) in industrial conditions can produce very different results.
Reviewer 3 Report
The manuscript '' Identification and Migration Studies of Photolytic Decomposition Products of UV-Photoinitiators in Food Packaging ''is interesting and fits into the scope of this journal. In addition, the authors present useful data related with photoinitiations and their degradation products, which has become a relevant topic in the field of food contact materials. The paper is well written and may be valuable for the scientific community. I recommend publishing the manuscript after minor revision. Specific comments- Introduction Line 83- there is mistype before NIAS- Please, remove. Results and discussion Lines 130-Please put ''tentatively identified'', unless you used standards to confirm their identity. Lines 225-228- It would be good to put a couple of examples in the text. Lines 245-266- Some of the tentatively identified products are commercially available- Have you injected real standards and prove the identity of the degradation products? Lines 331- CFR part 170-139-Please double check this. Material and methods Please include relevant parameter for the method validation (LOD, RSD, r2, etc) Have you used control-blanks in the experiments? Please include and make it clear in the text. What standards have you used to validate the GC and LC method? How can the authors be sure that they have captured most of the potential degradation products? Please explain. After these edits, the manuscript can be accepted for publication.Author Response
Please see the attachment.
